# Influence of Adipose-Derived Stromal Vascular Fraction on Resorption of a Large-Volume Free-Fat Transplant Evaluated Using T3D Optical Scanning

Matic Koren [1,*], Simona Kranjc Brezar [2], Tadej Dovšak [1], Gregor Sersa [2], Andrej Kansky [1] and Nataša Ihan Hren [1]

1   Department of Maxillofacial and Oral Surgery, University Medical Centre Ljubljana, 1000 Ljubljana, Slovenia
2   Department of Experimental Oncology, Institute of Oncology, 1000 Ljubljana, Slovenia
*   Correspondence: matic.koren@kclj.si; Tel.: +386-41-731-079

**Abstract:** Background and Objectives: The main drawback of lipofilling is fat transplant volume loss, which makes long-term results unreliable. This study's aim was to assess the influence of an adipose-derived stromal vascular fraction (SVF) on volume retention in large-volume fat grafts. Materials and Methods: A murine model was used for the in vivo evaluation of fat-graft volume changes over 6 months. We used 28 immunocompromised nude NU(NCr)-Foxn1nu mice and human fat tissue as a liposuction by-product. Part of the fat tissue was used for SVF preparation. We created a fat transplant without SVF (SVF-) and with SVF (SVF+) groups. Large-volume grafts were injected above the sacrum and scapula in the same animal. Volume loss was evaluated using three-dimensional optical scanning at 14 days (T1), 3 months (T2), and 6 months (T3) after transplantation. Scans were processed with Artec Studio software to obtain stereolithography files. The volumes were calculated in RapidForm software 2006. Results: The highest volume loss was observed above the scapula at T3 (SVF- 85%; SVF+ 75%). There was a significant difference in volume between SVF-/SVF+ for grafts above the sacrum at T2, with lower loss in SVF+, and the significance became stronger at T3. The difference in volume loss was also significant above the scapula between SVF-/SVF+ at T3. Conclusions: Although we found a beneficial effect of SVF on the long-term survival of large-volume fat tissue transplants, volume loss due to other contributing factors was high.

**Keywords:** lipofilling; murine model; stromal vascular fraction; free-fat transplant; optical scanning

## 1. Introduction

The use of autologous fat transplants is widespread through different medical disciplines for reconstructive and aesthetic purposes. Adipose transplants are readily available, easily obtainable with low donor-site morbidity, bio-compatible, and the procedure is inexpensive, repeatable, and versatile. The first attempt to transfer fat tissue was described in 1889 by Van der Muelen [1]. The techniques of harvesting, processing, and injecting adipose grafts were developed and modified throughout the 20th century. Currently, the most commonly used technique is one developed by Coleman [2,3]. To improve results for augmentations of large-volume defects, some modifications are recommended [4].

Autologous fat grafts are useful not only as fillers to correct volume defects but also for their regenerative potential [5,6]. Fat tissue is the biggest source of stem cells in the human body [7]. On the basis of its origin, these stem cells have been named adipose-derived stem cells (ASCs) [8,9]. In addition to the surgical technique used, these ASCs have a decisive role in promoting long-term fat transplant survival [10,11]. The long-term survival of transferred adipose tissue reportedly ranges between 25% [12–14] and 80% [4,15] of the initial volume. This unpredictable permanence could be due to variability in techniques, the graft's volume, or the site features in the recipient area. Fat transplant survival is inversely correlated to its volume [16]. Large-volume grafts have a higher resorption rate because of the long distance between the outer edge and the center of the transplant.

On the other hand, the survival of large-volume fat transplants could be improved with stem cell enrichment. A stromal vascular fraction (SVF) [17] or isolated and ex vivo expanded ASCs [10,18] can be used. An SVF is a cell fraction obtained from lipoaspirate with an enzymatic collagenase degradation of fat tissue. Enrichment with SVF or ASCs could improve the survival rate as high as 80% of the starting graft volume [18].

This study's aim was to evaluate the resorption of large adipose tissue transplants over time when enriched with an SVF in a murine model with nude mice. For the objective volume-change evaluation, we introduced an innovative method to our research protocol for volume measurement performed with three-dimensional optical scanning.

## 2. Materials and Methods

### 2.1. Animal Model

This study was conducted in a nude mouse model with fat tissue transplanted from a human donor. Given the need for animal housing, care, and appropriate laboratory conditions, the study was conducted at the Department for Experimental Oncology, Institute of Oncology, Ljubljana, Slovenia. Fat harvesting was performed at the Department of Maxillofacial and Oral Surgery, University Medical Center, Ljubljana, Slovenia. The experiment was approved by the Slovenian Ethics Committee for Animal Experimentation (No. U34401-5/2018/10) and The National Medical Ethics Committee (No. 0120-436/2017/4).

The sample comprised 28 immunodeficient and hairless nude NU(NCr)-Foxn1nu mice. Due to genetic modification, the mice were athymic and, consequently, without T-cell response, which makes them appropriate for tumors or xenograft biology studies. All mice in this study were 6-week-old females at the study start and were divided into two groups: the group without SVF (SVF-; $n$ =14) included mice that only received a fat transplant, and the group with SVF (SVF+; $n$ = 14) included mice that received SVF-enriched fat grafts. All mice lived under the same conditions with free access to fresh water and food. Five mice were housed in one individually ventilated cage. Weight was checked once per week from the day of the fat-graft application.

### 2.2. Fat Tissue Source

The fat tissue for this study was from the by-product of a planned liposuction for one human patient who signed consent to participate in the study. Using the technique introduced by Coleman, liposuction was performed under general anesthesia [3]. We used tumescent fluid. The total amount of needed lipoaspirate was approximately 100 mL, and, after harvesting, it was centrifuged at 3000 rpm for 3 min. Three layers were created: the top layer comprised oil from degraded adipocytes, the middle layer comprised fat tissue pieces, and the bottom layer comprised blood cells, water, and the injected mixture remains [3]. We obtained approximately 50 mL of fat tissue, and approximately 35 mL was used for SVF production. The rest of the volume was used as a fat graft.

### 2.3. Preparation of Fat Tissue for Transplantation

SVF was prepared in cooperation with Educell Company for Cell Biology Ltd. (Ljubljana, Slovenia). Approximately 35 mL of fat tissue was digested with 0.75 PZ U/mL of NB6 collagenase (Serva, Germany) at 37 °C for 30 min with shaking. The solution was then centrifuged, the cell pellet was transferred to a fresh vessel, and the cell suspension was rinsed twice with standard Ringer solution (B Braun, Germany). After the last rinse, the cell pellet was resuspended in Ringer solution to a volume of 3 mL. The whole cell count in suspension was $3.05 \times 10^6$/mL, and the ASC share was $0.11 \times 10^6$/mL.

We filled 56 syringes with 0.4 mL of fat tissue. In 28 syringes, we added 0.1 mL of Ringer lactate. In the remaining 28 syringes, 0.1 mL of SVF was added for fat-graft enrichment. The prepared mixtures were subcutaneously injected into the mice. Injection was performed under general inhalation anesthesia with 3% isoflurane. The mice were marked by ear holes for further identification. Grafts were injected as one bolus with a 23G needle in subcutaneous tissue above the scapula and sacrum (Figure 1). Each mouse

received the same composition of the mixture in both places. In SVF-, we injected the contents of 27 syringes containing a mixture of fat tissue and Ringer lactate. In SVF+, we injected the contents of 28 syringes containing fat tissue enriched with SVF. Fat tissue could not be implanted due to technical problems with syringes at two places in SVF- and at one place in SVF+.

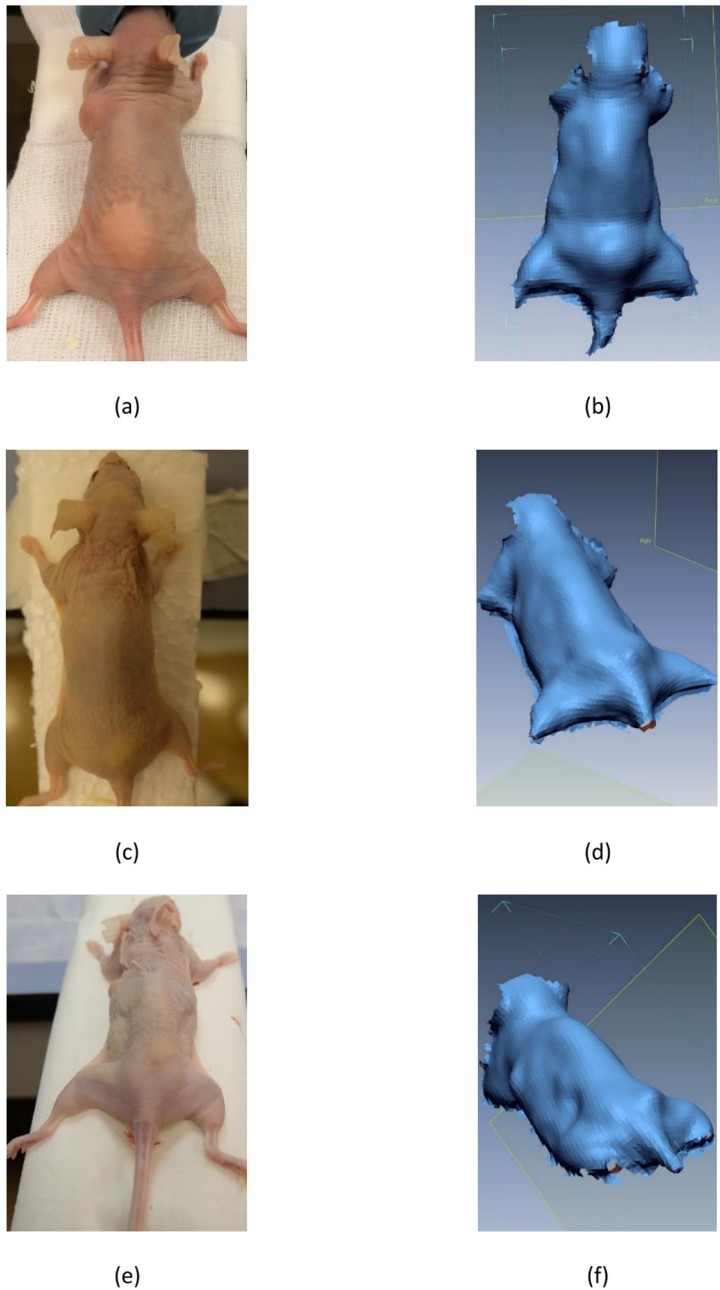

(a)

(b)

(c)

(d)

(e)

(f)

**Figure 1.** Mice photographs and scans at different time points: (**a**) photography at T1, (**b**) scan at T1, (**c**) photography at T2, (**d**) scan at T2, (**e**) photography at T3 and (**f**) scan at T3. T1, 14 days after fat graft application; T2, 3 months after fat graft application; T3, 6 months after fat graft application.

### 2.4. 3D Scanning

Before we injected the grafts, the mice were scanned using a 3D Artec Eva camera (Artec 3D; Senningerberg, Luxembourg). Each mouse was placed in a standard position. During the scanning, the tail and neck were in the same axis, and the legs were maximally stretched. New scans were performed in the second week (T1) after fat tissue application when the majority of the edema and hematomas had disappeared. Scanning was repeated

at 3 (T2) and 6 (T3) months after the fat injections. All scans were performed with the mice under general anesthesia in the described position.

The raw 3D scans were processed with Artec Studio 15 software to produce STL files. Further analysis was performed in RapidForm 2006 (Inus Technology Inc.; Rock Hill, SC, USA). We obtained images of the exterior surfaces of the mice bodies. Software was used to perform the best-fit procedure between the first scans and all later scans. For the best fit, we determined two points; the base of the tail and the occipital area of the head in the midline (Figure 2). Protuberances in the skin were observed where the fat grafts were injected, and the areas around them were trimmed. We obtained images of the exterior surfaces of the fat transplant. The volume in mm$^3$ was calculated with software. Each step from the best-fit process to volume calculation was repeated four times by the same examiner. The average value from measurements was used for further statistical analysis.

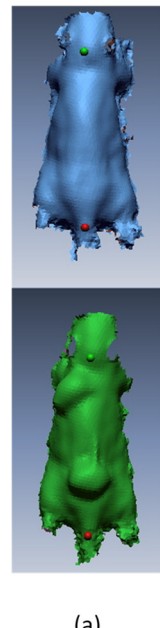 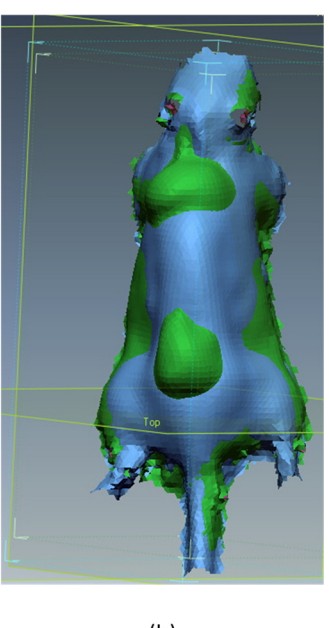

(a)                                                      (b)

**Figure 2.** Best fit on scan without fat transplant: (**a**) points used for best fit, (**b**) mouse model after best-fit procedure.

### 2.5. Statistical Analysis

Statistical analysis was performed in SPSS software (SPSS Inc., PASW Statistics for Windows, Version 18.0. Chicago, IL, USA). The mean and standard deviation were calculated for the fat graft volumes. The same procedure was also performed for the animal weight data to compare weight changes between groups. The independent Student's *t*-test was performed to determine the significance of differences between the groups at different time points. Statistical significance was set at $p < 0.05$.

## 3. Results

The mean volumes for the fat transplants above the left scapula and sacrum in both groups are presented in Table 1. In SVF-, the mean fat transplant volume at T1 above the scapula was $176.6 \pm 64.5$ mm$^3$, which decreased to $36.2 \pm 22.2$ mm$^3$ after 6 months. The fat tissue transplant mean volume at T1 above the scapula in SVF+ was $192.3 \pm 70.6$ mm$^3$ and was $79.6 \pm 35.7$ mm$^3$ 6 months later. The graft above the sacrum in SVF- at T1 had a mean volume of $240.6 \pm 93.7$ mm$^3$. At T3, the mean volume decreased to $36.7 \pm 33.0$ mm$^3$. In SVF+, the mean volume was $286.8 \pm 90.2$ mm$^3$ at the beginning and was $70.3 \pm 29.1$ after 6 months. The volume changes at T1, T2, and T3 of the transplant above the left scapula and sacrum in SVF- and SVF+ are visually presented in Figure 3.

**Table 1.** Comparison of the mean volumes of the grafted areas above the left scapula and sacrum and animal weights between the group SVF- and group SVF+ at T1, T2, and T3.

| | SVF- | | | SVF+ | | |
|---|---|---|---|---|---|---|
| | **T1** | **T2** | **T3** | **T1** | **T2** | **T3** |
| | Scapula region | | | | | |
| Mean (mm$^3$) | 176.6 | 77.5 | 36.2 | 192.3 | 101.8 | 79.6 |
| SD | 64.5 | 43.9 | 22.2 | 70.6 | 24.0 | 35.7 |
| n | 12 | 12 | 10 | 13 | 12 | 12 |
| | Sacrum region | | | | | |
| Mean (mm$^3$) | 240.6 | 56.8 | 36.7 | 286.8 | 96.1 | 70.3 |
| SD | 93.7 | 30.5 | 33.0 | 90.2 | 43.5 | 29.1 |
| n | 14 | 12 | 10 | 14 | 13 | 12 |
| | Animal weight | | | | | |
| Mean (g) | 26.1 | 27.7 | 28.1 | 25.4 | 27.3 | 26.0 |
| SD | 1.1 | 1.1 | 1.8 | 1.2 | 1.5 | 2.9 |
| n | 14 | 12 | 10 | 14 | 13 | 12 |

T1, 14 days after fat graft application; T2, 3 months after fat graft application; T3, 6 months after fat graft application; SVF-, control group; SVF+, group with added stromal vascular fraction; SD, standard deviation; n, number of subjects in sample.

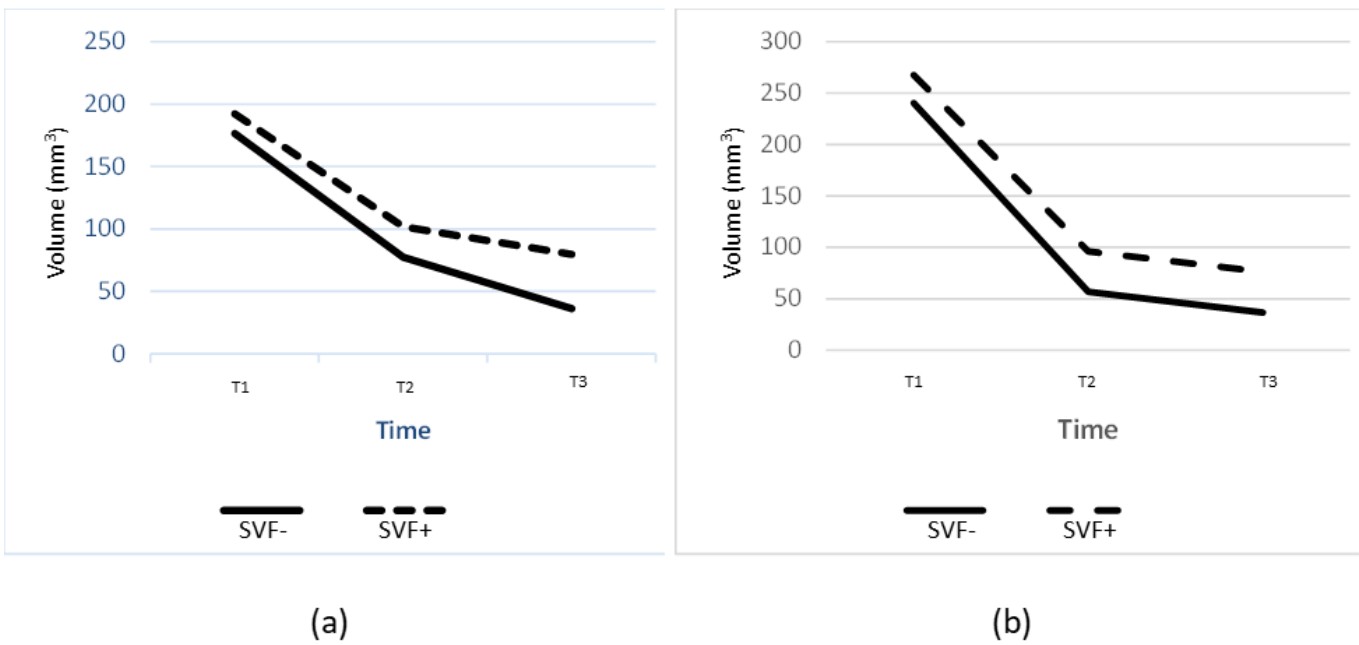

(a)                                        (b)

**Figure 3.** (**a**) Volume changes of fat transplant at T1, T2, and T3 above left scapula region in control group SVF- and SVF+. (**b**) Volume changes of fat transplant at T1, T2, and T3 above sacrum region in SVF- and SVF+; T0, 14 days after fat graft application; T2, 3 months after fat graft application; T3, 6 months after fat graft application; SVF-, control group; SVF+, a group with added stromal vascular fraction.

At 14 days after application, there was no significant difference between SVF- and SVF+ fat grafts volumes (Table 2). At 6 months, the fat transplant volume loss was significantly higher in SVF- than in SVF+ ($p < 0.05$).

**Table 2.** Comparison of volume changes above the left scapula and sacrum at different time points with 95% CI) between group SVF- and SVF+.

|  | *p*-Value SVF-:SVF+ | 95% CI | |
|---|---|---|---|
|  |  | Lower Border | Upper Border |
| Scapula region |  |  |  |
| T1 | 0.56 | −72.15 | 40.09 |
| T2 | 0.11 | −54.30 | 5.65 |
| T3 | 0.003 | −70.45 | −16.19 |
| Sacrum region |  |  |  |
| T1 | 0.2 | −117.76 | 25.29 |
| T2 | 0.01 | −68.87 | −9.69 |
| T3 | 0.02 | −59.86 | −7.23 |

T1, 14 days after fat graft application; T2, 3 months after fat graft application; T3, 6 months after fat graft application; SVF-, control group; SVF+, group with added stromal vascular fraction; CI, confidence interval; values of *p* < 0.05 were considered to be statistically significant.

The loss of sample animals in our study in the late period of the study (mean time at 18th week) was expected (6 of 28). In all cases, blood was present around the anus from which we concluded that the probable cause of death was gastrointestinal inflammation. The animal mean weights did not change significantly over time between the groups.

## 4. Discussion

In this study, we chose an animal model that enabled the study of the influence of SVF in large-volume fat transplants in a standardized environment. Too many patient-dependent variables would have complicated evaluations of the SVF effect on fat transplant survival in a clinical study. Harvesting of autologous mice fat tissue, usually inguinal fat pads, is limited in volume [19] and could be potentially associated with higher animal mortality due to the general anesthesia or postoperative wound inflammation. To provide enough adipose tissue for grafting, we decided to use human fat tissue, which was transplanted into immunodeficient nude mice.

In our study design, we decided to wait 14 days after fat transplantation before performing 3D scans of the mice. After this period, stable conditions inside and around the graft were expected due to resorption of the graft's liquid part and surrounding tissue swelling. Consequently, we chose more comparable starting points between the fat transplants. The later scans were performed at 3 and 6 months after the fat tissue application due to more stable graft volumes, as previously reported [20].

Adipose tissue was harvested using a Coleman technique. We chose the umbilic region as the donor site because of its accessibility. As recommended, we used 3 mm cannulas for fat harvesting [21]. Smaller fat particles are more beneficial for adipocytes survival because of nutrient diffusion [22]. Harvesting could also damage viable adipocytes because of shear stress. In the literature, larger diameter cannulas are recommended to minimize shear stress [23,24]. The influence of negative pressure, a local anesthetic, and tissue-refinement protocols were not considered in these studies. Smaller diameter multi-perforated cannulas improve the quality of harvested tissue. The count of ASC and vascular cells, both vital for neovascularization, is compared to the conventional higher technique [25].

Harvested fat tissue is a mixture of fat particles, blood cells, tumescent fluid, and oil. Different techniques are known to separate viable fat particles from blood and fluid. The sedimentation method is inferior to washing and centrifugation techniques [26]. In contrast, excessive centrifugation force could also damage adipocytes. Compared with other techniques, an appropriate centrifugation force could increase the ASC count and other growth factors in a fat transplant [27]. The blood cells and fluids in lipoaspirate could cause fat transplant degradation and an erroneous volume impression [28]. Our

protocol complied with Coleman's technique. The 3 min centrifugation gave the most favorable results [29]. Properly refined excess lipoaspirate added to the fat graft enhances fat transplant survival [30]. A previous animal study confirmed the superiority of adding ASCs to fat graft survival [31]. The number of ASCs is high in the liquid phase of a lipoaspirate, but liquid is usually discarded in the preparation process [32]. There are also differences in the number of cells between individual patients [10,18]. A certain number of ASCs are needed in fat transplants for successful lipofilling. It is also necessary to concentrate ASCs before the graft is implanted. In our study, we decided to use an SVF, which contained ASCs in addition to other progenitor cells. For clinical use, it is more appropriate to produce SVF because of the simplicity of the process and lower costs than those associated with the isolation of ASCs [33,34]. The SVF preparation from a lipoaspirate can be completed in a few hours, but isolation and culturing of ASCs could take weeks.

There are no reports describing a significant difference between the beneficial effects of ASCs on long-term fat transplant volume [35]. The effect of ASCs alone or as part of an SVF is somewhat controversial. Some reports have not found any difference in volume retention when using an SVF [36]. On the other hand, ASCs and an SVF have been found to have an important effect on fat graft retention [31,37].

Fat grafts have been injected into subcutaneous tissue above the scapula and sacrum of mice with a small-gauge needle, which causes less trauma to the recipient site and more stable conditions for fat graft survival [28]. A narrow-diameter cannula for injection could also damage fat particles because of shear stress [23], so the injection cannula should be carefully chosen considering these facts. At transplantation, mature adipocytes are endangered by hypoxia. The only way to provide the nutrients from the surroundings is after graft implantation, which occurs through diffusion [8]. Consequently, there are limitations to the reconstruction of large-volume defects. To obtain the desired result, it is necessary to repeat the procedure more times [38]. The retention rate after the second lipofilling is much higher [39], which also means higher costs and a risk for more complications.

Many theories have described the mechanism of adipose transplant survival. At present, the most accepted cellular survival theory is the one presented by Peer in 1955 [2]. A count of vital adipocytes at the time of transplantation has a crucial effect on transplant survival [40].

In our study, we used a 3D scanner to objectively evaluate changes in fat graft volume over time [41]. The 3D scanner uses flying triangulation to capture a 3D surface [42]. This method has very good reproducibility and accuracy for volume evaluation [43,44]. A previous comparison of two methods for volume measurement, computed tomography (CT), and 3D scanning showed overestimation of volume retention when CT scans were used [20]. Magnetic resonance imaging (MRI) and 3D scanning are the most reliable methods for evaluating the success of lipofilling. For long-term evaluation, the 3D scanner is more useful than MRI because of its accessibility and lower costs [45].

The quality of the free fat graft could be assessed with a histological exam. The structure of normal fat tissue and high vessel density has a positive relation to good clinical outcomes. Fat tissue necrosis and fibrosis are related to a lower volume of free fat graft [31,37].

After 6 months, fat graft volume losses were higher than expected, with worse results in G1. The mean volume retention was 20% above the scapula and 15% above the sacrum of the initial volume, but no significant difference in volume retention was observed between the grafts at the sacrum and scapula. A significant difference in graft volume change was observed between SVF- and SVF+, with volumes higher in SVF+, which received fat transplants with SVF. In SVF+, the graft volume retention was approximately 50% of the initial volume above the scapula and 25% above the sacrum. In the SVF+ group, the loss was higher in the region above the sacrum. This finding may be because underlying bone could cause higher skin tension above the fat graft, and conditions might have become more ischemic. Consequently, volume loss was higher. On the other hand, higher volume loss could be apparent due to a bigger tissue bulge during 3D scanning as a consequence of

thinner subcutaneous tissue and solid underlying bone. The difference in the grafts above the sacrum between SVF- and SVF+ was already significant at T2, but, at T3, the *p*-value was higher because of higher graft loss after 6 months in both groups. The effect of added SVF in SVF+ probably was not strong enough to balance the ischemic conditions caused by skin tension. A higher SVF share possibly would have a beneficial effect on fat graft volume retention.

An SVF includes mature and progenitor endothelial cells, pericytes, hematopoietic cells, preadipocytes, macrophages, and pluripotent ASCs. Some of these cells are precursors of specific cells; on the other hand, ASCs are pluripotent [46]. This feature is crucial for preserving fat transplant volume [35]. The ASCs are represented in a share from 3% to 9% [47,48]. The success rate is also dependent on the ratio between SVF and fat tissue in the transplant, with the ideal ratio being approximately 1:4 for SVF to the whole graft volume [49]. In our study, the SVF share was one quarter of the fat graft, and the ASCs were 3% of the total cell number in SVF. Big fat transplants probably need a higher ASC count in the SVF or a bigger volume of SVF to compensate for ischemic conditions in the sacrum region.

The fat graft volumes in our study were large considering the mice body weights and could be compared to breast augmentation with lipofilling because they showed almost a 50% loss of initial volume after 4 months [50]. The skin tension above a large-volume graft is higher, which worsens the ischemic conditions [51]. In the long term, the final volume is smaller and the quality of adipose tissue is lower [52], which could explain the higher volume loss in our study. To avoid the undesirable skin tension above large-volume fat grafts, the Brava technique was introduced and combined with lipofilling for breast reconstruction. Vacuum pressure is used to enlarge the skin around the recipient bed before fat tissue is transferred. Skin tension is minimized, and vascularization is maximized. The main downside of this method is patient cooperation, which is dependent on their motivation [53]. In our study, we did not use the Brava technique, but we obtained similar results after 6 months in group SVF+ above the scapula. Volume loss was approximately 50%. We attributed this result to the beneficial effect of SVF. The drugs used for hormonal active breast cancer, such as Tamoxifen, do not have influence on ASC proliferation and fat graft survival [54].

Methods for observing fat transplant survival described in the literature are very heterogenic [3], so it is difficult to compare them with our method. Animal model studies lasted from 4 weeks up to 3 months on average, and the initial volumes were much smaller [55], so their results are difficult to compare with our results.

Large-volume fat tissue transplants have a higher rate of volume loss. Compared with other animal model reports, initial graft volumes were smaller, and loss was consequently lower [33]. The periods of observation also vary in the literature. The reports with shorter observation times had higher volume retention [56,57]. The highest loss in our study was after 3 months, which is in accordance with other experiments [20]. The underlying mechanism is the resorption of fluid and non-viable cells [58]. A second lipofilling leads to higher volume retention. This could also explain the higher volume loss of our large fat grafts [59].

The distance for new vessel growth in large-volume fat grafts is longer, and skin tension above the graft is higher. It is logical to improve the process of angiogenesis and replacement of the lost adipocytes. SVF appears to be the right technically accessible choice. Higher concentrations of ASCs in our SVF sample might have provided better results in this study. However, this would increase costs because cell culture techniques are involved, so this option would not be as useful in everyday clinical practice.

## 5. Conclusions

In our study, the beneficial effect of an SVF on large-volume fat graft survival was demonstrated. The combination of an SVF and fat grafts has made lipofilling a more predictable surgical procedure. In future studies, we recommend the use fat grafts with



smaller volumes to minimize the influence of ischemia and to strengthen the SVF effect. Further research is also needed to investigate the influence of a different ASC count in the SVF on long-term fat graft survival.

**Author Contributions:** Conceptualization, M.K., S.K.B., T.D., G.S., A.K. and N.I.H.; methodology, M.K., T.D., G.S. and S.K.B.; software, M.K.; formal analysis, M.K.; investigation, M.K. and S.K.B.; writing—original draft preparation, M.K.; writing—review and editing, S.K.B., T.D., G.S., A.K. and N.I.H.; supervision, N.I.H. All authors have read and agreed to the published version of the manuscript.

**Funding:** Slovenian Research and Innovation Agency (No. P3-0374); University Medical Centre Ljubljana (No. TP20220050); Slovenian Research and Innovation Agency (P3-0003).

**Institutional Review Board Statement:** This study was conducted in accordance with the Declaration of Helsinki and approved by the National Medical Ethics Committee (No. 0120-436/2017/4, 26/10/2017). The animal study protocol was approved by the Slovenian Ethics Committee for Animal Experimentation (No. U34401-5/2018/10, 22/03/2018).

**Informed Consent Statement:** Informed consent was obtained from all subjects involved in the study.

**Data Availability Statement:** All supporting data and datasets generated for this study are available on request to the corresponding author.

**Acknowledgments:** We thank the Department of Experimental Oncology, Institute of Oncology, Ljubljana, Slovenia, for their help in the study design and care for the animals.

**Conflicts of Interest:** The authors declare no conflict of interest.

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
