# Peer review of "Influence of Adipose-Derived Stromal Vascular Fraction on Resorption of a Large-Volume Free-Fat Transplant Evaluated Using T3D Optical Scanning"

_applsci, doi:10.3390/app13127100_

Round 1
Reviewer 1 Report
THE MANUSCRIPT PRESENTS DATA AND INNOVATIVE APPROACHES IN THE USE OF ADIPOSE TISSUE.
However, some points should be clarified and/or completed.
The first point is related to the objective:
The study aim was to evaluate the resorption of large adipose tissue transplants over time when enriched with an SVF in a murine model with nude mice. For the objective 59 volume-change evaluation, we introduced an innovative method to our research protocol for volume measurement performed by three-dimensional optical scanning.
a) It is very important to know the number of cells in each injection, both of SVF and ASC.
b) Being an innovative method to measure the volume, it is essential to compare it with other evaluation methods.
It is mentioned that cell survival is inversely proportional to volume, then:
On what basis was it decided to work on these volumes and the relationship of SVF and ASC; the areas where the grafts were applied, and why was only the volume evaluated and not other additional parameters to correlate, at least at the end of the study?
How were the controls handled? Were they injected with Ringer's solution? How did they react with the injection volumes?
How were the controls handled? Were they injected with Ringer's solution? How did they react to the injection volumes?
Line 110
Explain: Fat tissue could not be implanted due to technical problems with syringes at two places in G1 and 111 at one place in G2.
It is necessary to incorporate validation of the method. Is this reported the first time it has been done?
How many samples? 33?
I think many experiments are needed to validate the method. With what data are the results compared?
Characteristics of the equipment, software, etc.
Author Response
Dear reviewer,
We counted the number of all cells (SVF) and ASC in suspension before mixing it with fat tissue. In our opinion, there should not be any difference in cell count between syringes.
Due to our technical limitations (we didn't have access to MRI or CT for this research), we decided to use just one method. Also, additional methods would expose animals to additional general anesthesia and could increase animal mortality.
In our study we want to evaluate the volume change, which is the main question in clinical practice. We use the 3D scanner because of its accessibility and also in clinical practice.
Due to animal welfare restrictions in Slovenia, the highest volume injected subcutaneously into mice is 0,5ml. In our study design we planned to observe large-volume fat grafts survival, consequently we want to inject the highest possible volume. Cells were suspended and concentrated in fluid. We want to use the smallest possible amount of fluid, because of later resorption. In the literature we found different information about added cells and fat tissue ratio. This was the basis of our decision. The grafts were applied above the sacrum and scapula, because of better visibility on 3D scan. These two places also have different characteristics (the skin is more mobile above the scapula, above sacrum ist was applied above the periosteum, above the scapula on the muscles) which could affect graft survival. In clinical practice, volume loss is the main problem. We decided to address only volume because of it's clinical significance.
28 mice were part of an experiment. 5 mice were just for monitoring of the living conditions of other animals in the experiment to exclude the influence of outside factors. They were counted by mistake. I apologize for this inaccuracy. Mistakes are corrected.
At the beginning of application, we had problems with fat graft injection due to high pressure in the syringe, consequetly some fat tissue was lost. Then we decided to use bigger needles. These three animals were also included in further research because every animal is valuable.
3D-scanning as a method was already used in literature for soft tissue change observation. Because of it's accessibility and simple use, we decided to use it in our studies.
28 mice were part of an experiment. 5 mice were just for monitoring of the living conditions of other animals in the experiment to exclude the influence of outside factors. They were counted by mistake. I apologize for this inaccuracy. Mistakes are corrected.
Data were compared to other studies with large-volume fat grafts.
With best regard,
Matic Koren
Reviewer 2 Report
Dear Authors,
I read with interest your work "Influence of Adipose-Derived Stromal Vascular Fraction on Resorption of a Large-Volume Free-Fat Transplant Evaluated by T3D Optical Scanning" that discuss an interesting topic in an animal model.
We find is a well conducted work indeed I think is important to include and cite the work of Klinger et al about the effect of Tamoxifen of fat graft intake and about the role of centrifugation in fat pro cessation. I think they are extremely pertinent for your work indeed
Author Response
Dear reviewer,
Suggested citations were included in text.
With best regards,
Matic Koren
Reviewer 3 Report
The authors describe an in-vivo study on immunodeficient mice using large volume human adipose cells complemented by SVF to improve volume retention.
The paper itself clearly planned, all sections are well distinguished and augmented by numerous colour figures, which help understand the imaging techniques used.
While the results and conclusions are not novel, they are based on solid statistical data and help to extend the knowledge on the causes of adipose cell resorption, which may be beneficial in the future.
I do have a few suggestions and comments for the authors:
- 33 mice are mentioned both in abstract and in material and methods, whereas both study groups consisted of 14 mice, which adds up to 28 in total. Please clarify.
- T0 taken at 14 days post surgery is slightly misleading. I suggest renaming the timepoints to T1, T2 and T3. Similarly, renaming G1 and G2 to e.g. SVF+ and SVF- could improve understanding
- I feel that the discussion section is too long and some sections could be shortened/removed without compromising the quality of the paper.
minor spelling and grammar errors which do not have an impact on the readability.
Author Response
28 mice were part of an experiment. 5 mice were just for monitoring of the living conditions of other animals in the experiment to exclude the influence of outside factors. They were counted by mistake. I apologize for this inaccuracy. Mistakes are corrected.
Timepoints and groups were renamed.
The discussion was shortened.
With best regards,
Matic Koren